# Reproducibility Study of "Discover-then-Name: Task-Agnostic Concept Bottlenecks via Automated Concept Discovery"

## Abstract

The DN-CBM framework proposed by Rao et al. represents a significant advancement in concept-based interpretability, leveraging Sparse Autoencoders (SAEs) for automatic concept discovery and naming. Our study successfully reproduces DN-CBM's core findings, confirming its ability to extract meaningful concepts while maintaining competitive classification performance across ImageNet, Places365, CIFAR-10, and CIFAR-100. Additionally, we validate DN-CBM's effectiveness in clustering semantically related concepts in the latent space, reinforcing its potential for interpretable machine learning. Beyond replication, our extensions provide deeper insights into DN-CBM's interpretability and robustness. We show that the discovered concepts are more concrete and less polysemantic, favoring monosemantic representations, and that polysemantic concepts have minimal impact on classification. Our intervention analysis on the Waterbirds100 dataset supports DN-CBM's interpretability, and a novel loss function improves classification accuracy by reducing reliance on spurious background cues. In addition, we show through a user study the advantages of the new loss function on the interpretable concept selection for CIFAR-10. While our automatic concept intervention method offers an alternative to manual interventions, human selection remains more effective. These findings affirm DN-CBM's validity and highlight opportunities for further refinement in interpretable deep learning.

## 1 Introduction

Deep learning models are increasingly applied across diverse domains, but their lack of interpretability remains a critical challenge. This problem has fueled interest in inherently interpretable methodologies that facilitate clearer, more comprehensible model explanations for human understanding (Böhle et al., 2022; Koh et al., 2020).

Concept Bottleneck Models (CBMs) (Achtibat et al., 2023; Kim et al., 2018; Koh et al., 2020; Oikarinen & Weng, 2023) present a promising strategy by introducing a human-interpretable concept space between feature extraction and classification. However, traditional CBMs need a labelled concept training dataset, leading to the rise of label-free LLM-based CBMs (Panousis et al., 2023; Yang et al., 2023), with the disadvantage of having to query expensive LLMs and whose faithfulness has been called into question (Margeloiu et al., 2021). The work by Rao et al. (Rao et al., 2024) addresses these limitations by introducing DN-CBMs, a type of CBM that employs Sparse Autoencoders (SAEs) to autonomously discover latent concepts. This approach reverses the conventional workflow, which typically involves defining concepts in advance and then applying them to the input data. Instead, it begins by uncovering the concepts in a language-agnostic manner and subsequently assigning them the closest matching term from a predefined vocabulary.

In this paper, we reproduce the findings of DN-CBM to validate its claims and evaluate its reproducibility. We focus on the use of SAEs for concept discovery and analyze the trade-off between interpretability and performance across ImageNet (Deng et al., 2009), Places365 (Zhou et al., 2018), CIFAR-10 (Krizhevsky, 2009), and CIFAR-100 (Krizhevsky, 2009).

This study focuses on DN-CBM due to its intrinsic task-agnostic nature, which distinguishes it from models such as CEIR Cui et al., 2023. In DN-CBM, the sparse autoencoder is trained once and remains fixed, thereby necessitating only the retraining of a linear probe for adapting to new tasks, making this model more computationally advantageous. Furthermore, DN-CBM employs a fixed (but possibly tunable) vocabulary based on unigrams rather than relying on expensive large language models for generating task-specific concepts. This design not only reduces computational overhead but also enhances reproducibility and ease of integration into various downstream applications. Additional advantages include the simplicity of the training procedure, good performances on complex datasets such as ImageNet and improved efficiency, which together render DN-CBM a compelling candidate for further empirical validation.

Additionally, we extend the research from Rao et al. by conducting a deeper exploration of the Sparse Autoencoder (SAE)'s concept space through various interventions. Specifically, we introduce three key extensions beyond the original work. First, we analyze the learned concept space using different vocabularies to assess its interpretability. To examine concreteness, we leverage the vocabulary developed by Brysbaert et al. (Brysbaert et al., 2014), which provides concreteness ratings for 37,058 English words and 2,896 two-word expressions based on sensory and motor experiences, predominantly visual and haptic. This enables us to evaluate the alignment between discovered concepts and human perceptual concreteness levels. To study polysemanticity, we utilize WordNet's hierarchical structure, identifying whether the learned concepts belong to multiple synsets. Second, we introduce a modified loss function designed to penalize spurious correlations, improving robustness by discouraging reliance on incidental features. Finally, we develop an automatic concept intervention method that selectively adjusts the influence of learned concepts without manual intervention. These extensions provide a more comprehensive understanding of DN-CBM's interpretability and its potential for refining concept-based representations.

## 2 Scope of reproducibility

The paper proposes a novel task-agnostic framework, DN-CBM, for constructing concept bottleneck models (CBMs). Below are the main claims made by the authors:

- **Claim 1: Task-Agnostic Concept Discovery**
  DN-CBM uses sparse autoencoders (SAEs) to discover concepts (Bricken et al., 2023) learned by CLIP (Radford et al., 2021a) in a task-agnostic manner, eliminating the need for labeled datasets or task-specific concept generation, making the approach computationally efficient and scalable.

- **Claim 2: SAE overcomes polysemanticity** DN-CBM uses sparse autoencoders (SAEs) to disentangle representations, ensuring each concept is mono-semantic and independently encoded (Elhage et al., 2022), improving interpretability.

- **Claim 3: Automated Concept Naming**
  DN-CBM automatically names the discovered concepts by aligning the dictionary vectors of the SAE decoder with the most similar text embeddings in the CLIP space. This naming mechanism is faithful to what the concept truly represent and avoids reliance on external large-language models.

- **Claim 4: Semantically Meaningful and Diverse Concepts**
  Sparse autoencoders enable the discovery of semantically meaningful concepts across different abstraction levels.

- **Claim 5: Semantic Consistency and Clustering in Concept Space**
  Semantically similar concepts and their associated images naturally cluster together in the latent concept space.

- **Claim 6: Intervention Analysis**
  Retaining only class related concepts (Petryk et al., 2022; Rao et al., 2022) significantly improves overall accuracy, while removing these concepts causes a substantial drop in performance, demonstrating the effectiveness of targeted interventions.

We structure our reproducibility study as follows: Section 3 details the models, datasets, and experimental setup used to replicate the original findings. Section 4 presents the reproduced results alongside the methods and results beyond the original paper. Finally, Section 5 summarizes our study and reflects on the challenges of reproducing the original work.

## 3 Methodology

To reproduce the results obtained by Rao et al. (Rao et al., 2024) we mainly used the code made publicly available by the authors in a GitHub repository [1]. The available code was useful for claims **1**, **3**, **4**, and **5**, while for the user study, the intervention analysis and additional experiments, some of which concerning the concreteness level of the vocabulary used for concept naming, we implemented additional scripts.

### 3.1 Model Descriptions

#### 3.1.1 DN-CBM model

The reproduced DN-CBM model introduces a structured approach to concept-based learning by organizing its pipeline into three main stages: (1) Concept Discovery with Sparse Autoencoders (SAE), (2) Automated Concept Naming, and (3) Concept Bottleneck Model (CBM) for Classification. Below we provide a high-level explanation of DN-CBMs, for a detailed explanation see Appendix 6.1.

**Stage 1: Concept Discovery with Sparse Autoencoders**   To extract human-interpretable concepts, we employ a Sparse Autoencoder trained on CLIP embeddings. Unlike conventional autoencoders that prioritize reconstruction accuracy, the SAE introduces a L1-sparsity constraint in the latent space, ensuring that individual neurons correspond to distinct, disentangled concepts (O'Neill et al., 2024; Rao et al., 2024).

The SAE consists of a linear encoder-decoder architecture, where the encoder projects CLIP embeddings into a high-dimensional sparse concept space, and the decoder reconstructs the original input. The training objective balances reconstruction accuracy and sparsity, ensuring that concepts remain disentangled while preserving key information from CLIP representations. The total number of trainable parameters of the SAE is of approx. 17 million for CLIP ResNet-50 and of approx. 8 million for CLIP ViT/B-16.

**Stage 2: Automated Concept Naming**   After concept discovery, we can now map the input data to a sparsely activated layer where each of the neurons represent a concept. To make the neurons of the hidden layer (concept space) human-interpretable we now assign them names from a predefined vocabulary $\mathcal{V}$. In particular, for every concept we compute its CLIP embedding (i.e. we pass the corresponding vector of the standard basis through the SAE decoder) and assign the name in $\mathcal{V}$ whose CLIP embedding vector is closest under cosine similarity. This ensures that each neuron is assigned the most semantically meaningful name, although the effectiveness of this step depends on the granularity of the vocabulary, since a richer vocabulary allows for more precise naming.

**Stage 3: Concept Bottleneck Model for Classification**   Once sparse representations are extracted and labeled, we use them to train a Concept Bottleneck Model (CBM) for classification. A linear probe is trained on top of the sparse concept representations to map them to class logits. The training objective consists of a cross-entropy loss for classification and a L1 regularization term to encourage sparse weight connections, further enhancing interpretability.

Crucially, both the CLIP encoder and SAE remain frozen during classifier training, ensuring that only the final classification layer is optimized. This enables direct alignment between image and text embeddings while maintaining the semantic structure of the learned concepts.

---

[1]https://github.com/neuroexplicit-saar/Discover-then-Name. Accessed 31/01/2025.

### 3.1.2 CLIP Image and Text Encoders

CLIP is a vision-language model that aligns images and text into a shared embedding space by training on large-scale image-caption pairs (OpenAI, 2021). It comprises an image encoder $\mathcal{I}$ and a text encoder $\mathcal{T}$, both of which map their respective inputs into the same multimodal space, facilitating zero-shot learning.

### 3.1.3 Concept Intervention on the Waterbirds100 Dataset

The concept intervention experiment follows the methodology proposed in the original paper and is conducted on the Waterbirds100 (Petryk et al., 2022; Sagawa* et al., 2020) dataset. This dataset is constructed such that all *landbirds* appear against *land* backgrounds and all *waterbirds* appear against *water* backgrounds in the training split, creating a strong spurious correlation between bird class and background. However, in the test split, this correlation is removed, making classification based on background cues ineffective.

We use the same components as the original study. The SAE trained on CC3M for the CLIP RN-50 model. The Linear Probe is trained for 200 epochs with a learning rate of 0.1 and a sparsity coefficient of 10.

After training, we extract the top 10 concepts with the largest weights in the probe and manually intervene by removing (zeroing out) concepts that are not bird-related. This differs from the original approach, where only the top 5 concepts were considered. The top 10 concepts for each class, sorted by weight, are presented in Table 8 in the Appendix 6.6 .

### 3.1.4 Extension: Penalizing spurious correlations with a modified loss on the Waterbirds100 Dataset

Deep learning models often exploit spurious correlations in training data, leading to poor generalization when these patterns do not hold in test settings. This issue is evident in datasets like Waterbirds100, where background features (e.g., land or water) strongly correlate with bird class during training but not in testing. Standard classifiers rely on these cues, resulting in suboptimal performance when they become unreliable. Waterbirds100 is explicitly designed to introduce such correlations in training while removing them in testing, making it an ideal benchmark for robustness evaluation.

To address this, we introduce a modified loss function that penalizes reliance on spurious correlations. This function encourages the model to prioritize semantically meaningful concepts while suppressing irrelevant or misleading features. By leveraging class names and concept embeddings, we quantify semantic alignment and adjust model training accordingly.

**Metadata Usage**  Beyond training data $(\text{train\_X}, \text{train\_y})$, we incorporate class names (e.g., *waterbird*, *landbird*) to guide concept selection. Specifically, we extract a large set of candidate concepts and estimate their semantic alignment with each class. This produces a **similarity matrix** ($\#\text{classes} \times \#\text{concepts}$), where each row represents a class and each column a concept. A corresponding **related masks** matrix of the same size identifies key class-associated concepts. Similarity is computed by embedding class names via the CLIP text encoder and measuring alignment with CLIP's discovered concept embeddings.

By leveraging class metadata, our penalty loss reduces spurious correlations between backgrounds and bird types, improving generalization when background cues change in the test split. This ensures the model focuses on bird-relevant concepts rather than background biases.

**Experiments**  To evaluate the effectiveness of our penalty loss, we trained a linear probe on the Waterbirds100 dataset, with a learning rate of 0.1, a sparsity loss coefficient of 10, a penalty loss coefficient of $5 \times 10^{-5}$, over 300 epochs. For a more detailed mathematical explanation of the loss, see Appendix 6.2.

### 3.1.5 Extension: Automated Concept Intervention

Motivated by the need for a more scalable and practical approach, we introduced automatic concept intervention as an alternative to the manual intervention strategy proposed in the original paper on the Waterbirds-100 datasets. This automation allows the method to generalize more effectively to datasets with a large number of classes, where manual intervention would be infeasible.

This method aims to identify concepts that are semantically related to class names based solely on the embeddings of their textual representations. Unlike the modified loss function, which relies on the original learned concept embeddings (the dictionary vectors of the SAE Decoder), this approach directly utilizes the embeddings of concept names (from CLIP). For more detail see Appendix 6.2. This methodology is conceptually closer to the manual intervention performed in the original paper, allowing for greater flexibility in the choice of embeddings and enabling experimentation with different models, including: **BERT** (Devlin et al., 2019), **GloVe** (Pennington et al., 2014), **CLIP**, and **Sentence Transformer** (Reimers & Gurevych, 2019). The automatic concept intervention procedure consists of the following steps:

1. Train a linear probe. We evaluate this method using both a standard probe and a probe trained with the modified loss function.

2. Extract the top-$k$ concepts with the largest weights in the trained probe.

3. Zero out all remaining weights, retaining only the top-$k$ concepts.

4. Encode the class names and concept names using one of the four embedding models listed above.

5. Compute the similarity between the class name embeddings and the embeddings of the retained top-$k$ concepts, using a predefined similarity metric and a threshold.

6. Zero out the concepts whose similarity scores fall below the threshold.

The image encoder is implemented using two architectures: ResNet-50 (RN50) (He et al., 2015) and Vision Transformer (ViT-B/16) (Dosovitskiy et al., 2021). RN50 extracts hierarchical image features using convolutional layers, producing compact embeddings, while ViT-B/16 treats images as sequences of $16 \times 16$ pixel patches, modeling long-range dependencies through self-attention. Both architectures generate embeddings of size 1024 (RN50) and 512 (ViT-B/16). Pre-training on 400M image-text pairs using a contrastive loss function ensures that semantically similar images and text are pulled together in the embedding space.

### 3.2 Datasets

**Training the SAE**

• **Conceptual Captions 3M (CC3M) (Sharma et al., 2018).** Contains 3,318,333 training and 15,840 validation image-caption pairs. Images are resized to 224x224, normalized with CLIP and augmented[2].

**Training the Linear Probe**

• **ImageNet-1000 (Deng et al., 2009).** Contains 1,153,050 training, 128,117 validation, and 50,000 test images across 1,000 classes. Images are resized to 224x224, normalized and augmented[3].

• **Places365 (Zhou et al., 2018).** Contains 1,623,114 training, 180,346 validation, and 36,500 test images across 365 categories. Images are resized to 256x256, cropped to 224x224, normalized and augmented[4].

• **CIFAR-10 and CIFAR-100 (Krizhevsky, 2009).** Both datasets contain 45,000 training, 5,000 validation, 10,000 tests images. Images are standardized, with random cropping and horizontal flipping applied[5].

**Vocabularies**

• **20K Vocabulary (Oikarinen & Weng, 2023).** Contains 20,000 common English words. Words are tokenized, stemmed, stop words removed, and encoded[6].

---

[2] Available in https://ai.google.com/research/ConceptualCaptions/. Accessed 31/01/2025.

[3] Available with academic license in https://www.image-net.org/. Accessed 31/01/2025.

[4] Available in http://places2.csail.mit.edu/. Accessed 31/01/2025.

[5] Available in https://www.cs.toronto.edu/~kriz/cifar.html. Accessed 31/01/2025.

[6] Available in https://github.com/first20hours/google-10000-english/blob/master/20k.txt. Accessed 31/01/2025.

- **Concreteness Ratings (Brysbaert et al., 2014).** Contains concreteness ratings for 37,058 English words and 2,896 two-word expressions rated in an abstract (1) to concrete (5) scale[7].

- **WordNet Vocabulary (Fellbaum, (1998, ed.)).** Contains over 155,000 words hierarchized into synonym sets (synsets). Words are tokenized, stemmed, encoded, and hierarchical relationships are represented as directed acyclic graphs[8].

### 3.3 Hyperparameters

The hyperparameters for training the SAE and linear probes were predefined in the dictionary found in `dncbm/method_utils.py`. Consequently, no hyperparameter search was conducted for either model. The exact values used can be found in Table 1 and Table 2.

| Encoder | LR | L1 coeff. ($\lambda_1$) | Exp. factor ($\mu$) | Epochs | Resample freq. | Batch Size |
|---|---|---|---|---|---|---|
| CLIP ResNet-50 | 5e-4 | 3e-5 | 8 | 200 | 10 | 4096 |
| CLIP ViT-B/16 | 5e-4 | 3e-5 | 8 | 200 | 10 | 4096 |

Table 1: Hyperparameters used for the training of the SAE, for the encoders CLIP ResNet-50 and CLIP ViT-B/16. LR stands for Learning Rate and note that a expansion factor of 8 implies that the concept space will have 8192 dimensions for ResNet-50 and 4096 for ViT-B/16.

| Dataset | LR | L1 coeff. ($\lambda_2$) | Epochs | Batch Size |
|---|---|---|---|---|
| CIFAR10 | 1e-3 | 1 | 200 | 512 |
| CIFAR100 | 1e-2 | 1 | 200 | 512 |
| Places365 | 1e-3 | 1 | 200 | 512 |
| ImageNet | 1e-3 | 1 | 200 | 512 |
| Waterbirds-100 | 1e-1 | 10 | 200 | 512 |

Table 2: Hyperparameters used for the training of the linear probes. The values are the same for both encoders (CLIP ResNet-50 and CLIP ViT-B/16). LR stands for Learning Rate.

### 3.4 Experimental setup

During this study, we conducted experiments to reproduce the claims presented in the DN-BCM paper. For the core claims, the authors' provided codebase was sufficient. However, for additional claims discussed in Appendix 6.3–6.5, such as the user survey and concept intervention (claim **6**) on the Waterbirds100 dataset, we implemented supplementary scripts, which are available in our repository.

Beyond replication, we extended the authors' claim **2** by conducting auxiliary experiments to better understand the SAE's learned concept space. These experiments include using other vocabularies, analyzing the concreteness level of learned concepts, and investigating the impact of polysemantic words as concepts (see Section 4.2.1). Further experiments were performed to penalize spurious correlations with a modified loss (Section 3.1.4) and to develop an automatic concept intervention (Section 3.1.5).

All experiments were implemented using PyTorch (Paszke et al., 2019). We leveraged pre-trained feature extractors from the official CLIP repository, specifically ResNet-50 and ViT-B/16 models. The extracted features, after pooling, were used to discover concepts via a sparse autoencoder. We trained our sparse autoencoders following the methodology described by (Bricken et al., 2023), using their publicly available implementation[9] (v1.3.0).

---

[7] Available in https://link.springer.com/article/10.3758/s13428-013-0403-5. Accessed 31/01/2025.

[8] Available in https://wordnet.princeton.edu/. Accessed 31/01/2025.

[9] https://github.com/ai-safety-foundation/sparse_autoencoder. Accessed 31/01/2025.

For classification, we trained linear probes on the learned concept representations using the Adam optimizer (Kingma & Ba, 2017), without a bias term. The classifier was optimized with a cross-entropy loss function and an L1 sparsity constraint on the weights. We trained linear probes on all learned SAEs and selected the models that achieved the highest top-1 validation accuracy for each dataset.

### 3.5 Computational Requirements

Our experiments were conducted on the Snellius supercomputer using NVIDIA A100 SXM4 80GB GPUs (400W TDP) within an HGX A100 node architecture. Although each node includes four GPUs, our jobs utilized only a single GPU per node. Across all phases of this reproduction study, including the extensions, a total of 97 node hours were consumed.

To estimate the carbon footprint, we used the Machine Learning Impact calculator (Lacoste et al., 2019), which takes into account hardware specifications, runtime, and the carbon intensity. Given a reported carbon efficiency of 2.12 kgCO$_2$eq/kWh for Snellius, the total emissions were initially estimated at 82.26 kgCO$_2$eq. However, since only one out of four GPUs was actively used per node, we scale this estimate by a factor of 0.25, resulting in a corrected carbon footprint of approximately 20.57 kgCO$_2$eq.

## 4 Results

Our reproducibility study reveals that overall, all the claims mentioned by the authors are correct and reproducible. In this section, we first highlight the results reproduced to support the main claims of the authors, then we delve into additional results observed by studying more in depth the concept space.

### 4.1 Results reproducing original paper

### 4.1.1 Concept Discovery and Naming

We were able to reproduce the claims **3** and **4** of the authors regarding the quality of the extracted concepts and their names, the automatically chosen names for the concepts very often reflected the common feature in the concept images despite coming from very different datasets. Examples of our results can be seen in the Appendix 6.3. Furthermore, as suggested by the authors, we used a wider vocabulary for improving the concept naming, the results of this approach are shown in Section 4.2.1. To evaluate the quantitative aspect of the concept consistency and naming accuracy, we reproduced the user study proposed by the authors, and our trends reflected those shown by the authors (see Appendix 6.4 for details).

### 4.1.2 Clustering Concept Vectors

To further measure semantic consistency, we also reproduced the clustering evaluation performed by the authors, in order to represent how well semantically related concepts cluster together in the latent concept space. Our results aligned with the authors' claim **5**, showing that similar concepts, such as winter-related terms, formed coherent clusters. The reproduced clusters maintained strong semantic consistency, demonstrating that the concept-based representations effectively capture meaningful relationships. This reinforces the validity of the learned latent space and its ability to organize interpretable concepts. For more details, see Appendix 6.3.

### 4.1.3 Interpretability of DN-CBM

The classification performance of the authors' DN-CBM models, trained on ImageNet, CIFAR-10, CIFAR-100, and Places365, was accurately reproducible and their results often outperformed the baseline methods, as shown in the article. A summary of the reproduction analysis using CLIP ResNet-50 and CLIP ViT-B/16 is shown in Table 3. The interpretability assessments, detailed in Appendix 6.3, showed that our reproduced DN-CBM provided intuitive and class-relevant local and global explanations, aligning with prior findings. Furthermore, our results confirmed that the reproduced models provided meaningful explanations despite using a task-agnostic concept set, compared to the linear probe and zero-shot performance of the CLIP

model. These results validate the robustness of DN-CBM in both accuracy and interpretability, supporting the authors' claim **1**.

| Model | Task Agnostic | CLIP ResNet-50 | | | | CLIP ViT-B/16 | | | |
|---|---|---|---|---|---|---|---|---|---|
| | | IMN | Places | CIF10 | CIF100 | IMN | Places | CIF10 | CIF100 |
| Linear Probe | - | 73.3* | 53.4 | 88.7* | 70.3* | 80.2* | 55.1 | 96.2* | 83.1* |
| Zero Shot | - | 59.6* | 38.7 | 75.6* | 41.6* | 68.6* | 41.2 | 91.6* | 68.7* |
| DN-CBM (Rao et al.) | ✓ | 72.9 | 53.5 | 87.6 | 67.5 | 79.5 | 55.1 | 96.0 | 82.1 |
| DN-CBM (Reproduction) | ✓ | 72.7 | 53.1 | 86.7 | 69.1 | 79.6 | 54.9 | 96.0 | 82.4 |

Table 3: Performance of our CBM in comparison to the authors' work. The classification accuracy (%) of the CBM and baselines is achieved using CLIP ResNet-50 and ViT- B/16 feature extractors on ImageNet, Places365, CIFAR10, and CIFAR100. '*' indicates results reported for the respective baselines, zero-shot performance is as reported by (Radford et al., 2021b) and for Places365 we reported the values reported by the authors.

### 4.1.4 Concept Intervention on the Waterbirds100 Dataset

We were able to reproduce the concept intervention experiment as it was done in the original paper. The results are presented in Table 4, where performance changes relative to the standard probe are indicated in parentheses.

Table 4: Performance comparison across different intervention strategies. Relative changes for our results are shown in parentheses.

| Model | Overall | Worst Groups | | Training Groups | |
|---|---|---|---|---|---|
| | | L.Bird@W | W.Bird@L | L.Bird@L | W.Bird@W |
| *Original Paper's Results* | | | | | |
| Before Intervention | 82.8 | 71.3 | 57.5 | 98.6 | 93.3 |
| Only Bird Concepts | 89.4 (+6.6) | 86.6 (+15.3) | 71.3 (+13.8) | 96.8 (-1.8) | 91.4 (-1.9) |
| No Bird Concepts | 60.8 (-22.0) | 28.5 (-42.8) | 28.8 (-28.7) | 95.0 (-3.6) | 85.8 (-7.5) |
| *Reproduced Results* | | | | | |
| Before Intervention | 83.32 | 73.01 | 47.51 | 99.02 | 91.92 |
| Bird-Only Probe | 89.99 (+6.67) | 91.44 (+18.43) | 68.94 (+21.43) | 97.12 (-1.9) | 85.09 (-6.83) |
| No-Bird Probe | 56.29 (-27.03) | 15.90 (-57.11) | 18.01 (-29.05) | 98.22 (-0.8) | 94.40 (+2.48) |

The results indicate that removing non-bird-related concepts significantly improves performance on the hardest subgroups (W@L and L@W). The Bird-Only Probe outperforms the Standard Probe by a substantial margin on these groups, demonstrating that forcing the model to rely only on bird-related concepts mitigates the reliance on background information. In contrast, the No-Bird Probe performs significantly worse, particularly on W@L and L@W, confirming that bird-related concepts are essential for classification.

Notably, the Bird-Only Probe experiences a slight drop in performance on the training-aligned groups (W@W and L@L), suggesting that some non-bird-related concepts may have contributed positively to classification in these cases where the background is informative. We therefore accept claim **6** given the empirical results we obtained.

### 4.2 Results beyond original paper

#### 4.2.1 Understanding the learned concept space using different vocabularies

We performed an analysis of the concept space, for which we introduce two vocabularies that have additional statistics associated to each of the words: Concreteness ratings (Brysbaert et al., 2014) and WordNet (Fellbaum, (1998, ed.)). The former vocabulary consists of 40,000 words with a measure of concreteness between 1 (abstract) and 5 (concrete). The latter is a well known lexical database consisting of 155,000 words from which we extracted the average number of synsets (synonym sets) for every word. The motivation behind the study is to better understand the learned concept space as well as addressing two claims made by the authors regarding human-interpretability and polysemanticity of concepts. The results and data used for this section is available for open access[10].

**On the concreteness level of the learned concepts**

The authors argue that there is no assurance that the concepts learned by neurons correspond to human-interpretable concepts. If such non-interpretable concepts occur too frequently, they could hinder the interpretability of DN-CBMs. To examine this, we assess human interpretability of concepts on a scale ranging from abstract to concrete based on their assigned names. In Figure 1 we can observe that even though the vocabulary contains more abstract words, the learned concepts by the SAE are strongly distributed towards the more concrete ratings. Moreover, given that there is little correlation between concreteness and concept alignment (Figure 10 of Appendix 6.5) we can conclude that concepts learned by DN-CBMs will be mostly human-interpretable, given the tendency of the SAE to learn concrete concepts. This finding highlights the advantage of language-agnostic concept discovery over methods that predefine concept names, and emphasizes the need to take into account the discrepancy in abstraction levels between vision and language.

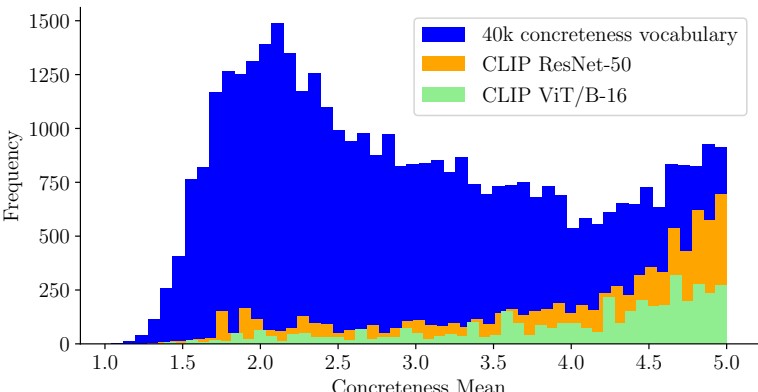

Figure 1: Distribution of concreteness of words for the 40k words with concreteness ratings vocabulary, the named concepts for both CLIP ResNet-50 and CLIP ViT/B-16. We can observe that despite abstract words being more frequent in the vocabulary, concrete words are more frequent in the learned concept space.

**On the effect of polysemanticity in classification**

The authors highlight the capability of SAEs in addressing the issue of polysemanticity with claim **2**, as supported by (Bricken et al., 2023). However, it is possible that the linear probe overly depends on these concepts, as they are activated for a broader range of inputs, which would ultimately undermine interpretability. We employ WordNet to analyze the presence of polysemantic concepts learned by the SAE and the effect these have in classification. In particular, we define a word as polysemantic if it belongs to more than one

---

[10]https://anonymous.4open.science/r/FACT-4635/FACT_Reproducibility_DNCBM.ipynb. Accessed 31/01/2025.

synonym set and monosemantic if it belongs to only one. Under this definition, more than 21% of lemmas in WordNet are polysemantic, whereas fewer than 13% of the named concepts learned using CLIP ResNet-50 are polysemantic, showing that the SAE prioritized monosemantic concepts.

In order to study the effect of polysemantic words in classification, we ablate polysemantic concepts and evaluate the trained linear probes on the test split of the four probe datasets. Table 5 shows that removing polysemantic neurons leads to only a minor decrease in accuracy, indicating their limited impact in overall performance. We also include the weight that polysemantic concepts have in concept activations, as a measure of how polysemantic the images in the dataset are. ImageNet is a less polysemantic dataset and thus the effect of polysemantic concepts in classification is less. In contrast, Places365 appears to be more polysemantic, leading to a greater drop in accuracy. We acknowledge that while CIFAR-100 aligns with this trend, CIFAR-10 does not, but we assume that the resolution of the images in the CIFAR datasets is too low to draw meaningful conclusions.

|  | ImageNet | Places365 | CIFAR10 | CIFAR100 |
|---|---|---|---|---|
| **% activation mass polysemantic** | 13.9 | 15.2 | 16.3 | 16.3 |
| **Accuracy before pruning** | 72.7* | 53.1* | 86.7* | 69.1* |
| **Accuracy after pruning** | 70.6 (-2.1) | 49.5 (-3.6) | 85.8 (-0.9) | 65.1 (-4.0) |

Table 5: Accuracies in classification of test split for CLIP ResNet-50 before and after pruning polysemantic concepts. The percentage of the mass activation coming from polysemantic concepts is first shown as a reference on how polysemantic the dataset is. This value is obtained by computing the mean activation of all concepts and then the percentage coming only from polysemantic concepts. '*' as reported by Table 3.

Lastly, we want to point out that even though polysemanticity doesn't have a big impact on classification, it is fundamental for the reconstruction process of the SAE. Given the sparsity condition, polysemantic words are more versatile and can be used in a larger range of input images. We named the concepts every three epochs during the training of the SAE and discovered that polysemanticity increases throughout training (Figure 11 of Appendix 6.5). This is likely due to CLIP embeddings of polysemantic words being concentrated in more densely populated regions and at the beginning of the training most of the random concepts are best aligned to monosemantic words.

### 4.2.2 Penalizing spurious correlations with a modified loss

Our final model achieved an accuracy of 88.75%, marking a relative improvement of +5.43% over our standard linear classifier on the Waterbirds-100. Unlike concept intervention, which requires manual human oversight to correct spurious correlations, our approach achieves this improvement in a fully automated manner. This result highlights the effectiveness of our penalty loss function in reducing reliance on background cues and guiding the model toward class-relevant features. With the scope of generalizing our results to a different dataset, we also trained the linear probe on CIFAR10 on the modified loss. The accuracy dropped slightly (87.95% and 86.59% for the original and modified loss, respectively; a decrease of -1.36%), but we noticed that the top activated concepts were better after applying the modified loss. In order to highlight the improvement of the concepts we conducted a user study, to better understand whether the potentially perceived increase in interpretability of highly activated concepts justifies the drop in accuracy.

**User study on CIFAR10 concept names' interpretability**

For each of the 10 classes of the CIFAR10 dataset, we analyzed the top 10 activated concepts, both before and after applying the modified loss, and for each class we selected the first three concepts appearing uniquely in the two concept lists (we ignored the concepts appearing in both lists). Each subject had to answer to two questions for each of the ten classes: the first was asking whether they found the concepts originated from the modified loss, over the other concepts, more related to the class name; the second question asked to rate individually all the six concepts (3 from the modified loss, 3 from the original loss) as not related,

somehow related or very related to the class name. In this way we were able to collect both qualitative and consistent results, presented in Figure 2. The survey was published internally and it received 52 responses.

In Figure 2a we can observe that the new loss produces highly activated concepts that participants find at least somewhat relevant, rather than completely unrelated, as happened most of the cases with the original loss. The most highly activated concepts given by the new loss (Figure 2b), the new loss was chosen over the original loss the majority of the time across different categories, indicating a strong overall preference for the outputs produced by the modified approach. Together, these results imply that the new loss generates concept relationships that users perceive as more appropriate or closer in meaning to the class label, hence showing that the slight drop in accuracy can be justified by a perceived increase in the interpretability of the linear probe.

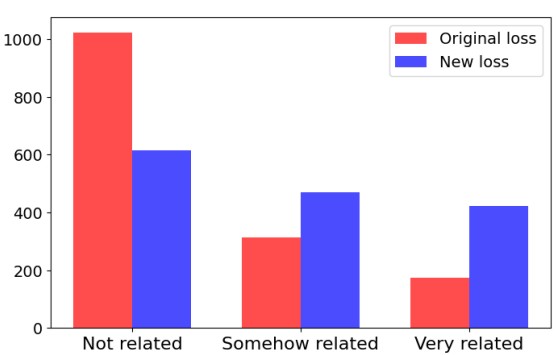
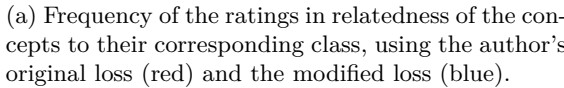

(a) Frequency of the ratings in relatedness of the concepts to their corresponding class, using the author's original loss (red) and the modified loss (blue).

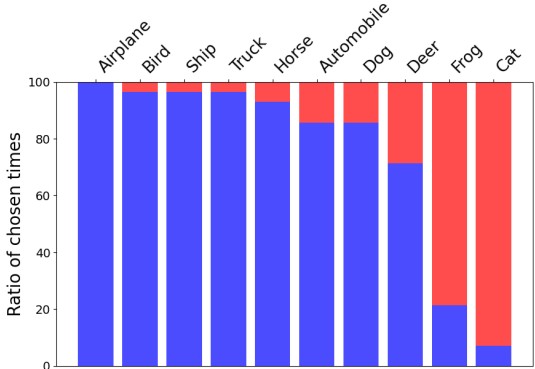

(b) The proportion of times survey respondents chose the concepts given by the new loss (blue) over the original loss (red), for each class of the CIFAR10 dataset.

Figure 2: Results of the user study to compare the interpretability of the concepts yielded with the original and the newly proposed loss for the linear probe. The study was conducted using the CIFAR10 dataset.

### 4.2.3 Automated Concept Intervention

Each of the embedding models were tested with multiple threshold values. However, our findings indicate that all automatic intervention methods underperform when compared to manual human intervention, which provides a more accurate selection of relevant concepts (see Table 6). This suggests that while embedding-based similarity metrics can serve as a useful heuristic, they may not fully capture the nuanced relationships between concepts and class semantics in a way that matches human intuition. As observed in Table 7, the selected concepts exhibit limited intuitive alignment with the class names and lack a clear semantic association with bird-related concepts.

Table 6: Test accuracies with automatic concept intervention. The relative performances (marked with '*') are compared to the base model without any concept intervention.

| Model | Test Accuracy |
|---|---|
| Sentence Transformer | 78.56 (-4.76*) |
| GloVe | 76.81 (-6.51*) |
| CLIP | 83.98 (+0.68*) |
| BERT | 68.97 (-19.35*) |

Table 7: Top-5 concepts for each class in the Waterbirds-100 using CLIP embeddings.

| Class | Concept 1 | Concept 2 | Concept 3 | Concept 4 | Concept 5 |
|-------|-----------|-----------|-----------|-----------|-----------|
| Landbird | Training | Oldham | Squash | Carlisle | Nightlife |
| Waterbird | Training | Oldham | Nightlife | Ireland | Leaf |

## 5  Discussion

Our reproduction study supports the claims made in the original paper. [11]. The core experimental results, such as the observed performance as compared to other CBM methods align well with the original findings, reinforcing the validity of the authors' claims. Our study successfully reproduced key findings of DN-CBM, validating its ability to extract meaningful concepts (Claim 3 and Claim 4) and maintain competitive classification performance across multiple datasets, including ImageNet, Places365, CIFAR-10, and CIFAR-100 (Claim 1). Additionally, our results confirm the claim that DN-CBM effectively clusters semantically related concepts in the latent space (Claim 5), reinforcing its potential for interpretable machine learning. Finally, our reproduction of the Intervention Analysis with the Waterbirds100 dataset further highlights DN-CBM's interpretability advantages (Claim 6).

Beyond replication, our extensions provide deeper insights into DN-CBM's interpretability and robustness, namely through the study of the effect of concreteness and polysemanticity in the learned concepts. We also introduced a novel loss function that penalizes reliance on spurious background cues, improving classification accuracy by encouraging the model to prioritize class-relevant concepts. This automated approach outperformed the standard linear probe and produced more interpretable results, demonstrating that DN-CBM's concept representations can be refined to mitigate biases. Additionally, our automatic concept intervention method, which selects influential concepts based on their alignment with class names, provides an alternative to manual interventions, though it remains less effective than human-guided selection.

Despite these advancements, DN-CBM is not without limitations. While concept discovery is task-agnostic, the extend of interpretability of the concepts remains dependent on the vocabulary used for alignment, necessitating further exploration of alternative naming strategies. Additionally, while DN-CBM successfully disentangles representations, certain abstract or existent polysemantic concepts may still lack clear human interpretability. Moreover, automated concept naming can introduce bias: while WordNet provides a relatively neutral vocabulary, models like CLIP may reflect societal or cultural biases present in their training data. This should be taken into account when interpreting concept labels.

### 5.1  What was easy

The original paper contains extensive research on the presented method as well as the methodology followed for the experiment. The provided code was functional and required minimal changes when used in GPU-based architectures. The hyperparameters chosen for the training of the SAE were documented in the paper and the weights obtained by the original authors were publicly available hence reproduction and comparing results was easy.

### 5.2  What was difficult

Several challenges emerged during the reproduction process. Using the provided code in CPU-based architectures or with no CUDA availability was deemed impossible, after many tries in different computers. To this regard, the code does not generalize to computing in CPU.

Additionally, while the majority of the experimental setup was clearly documented, the chosen hyperparameters for training the linear probes on the different datasets were not properly stated in the article which initially lead to confusion and waste of compute. However, we later found them in the code.

---

[11]Additional code implemented by us can be found at https://anonymous.4open.science/r/FACT-4635/README.md. Accessed 31/01/2025.

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

# 6 Appendix

## 6.1 Model technicalities

### 6.1.1 Concept Discovery with Sparse Autoencoders.

The sparse autoencoder operates on feature representations derived from a pre-trained CLIP model, which we dicussed in more detail in Section 3.1.2. The SAE consists of a linear encoder $f : \mathbb{R}^d \to \mathbb{R}^h$, which projects CLIP embeddings into a sparse concept space, and a decoder $g : \mathbb{R}^h \to \mathbb{R}^d$, which reconstructs the original embeddings. The encoder and decoder are parameterized as follows: the encoder uses a weight matrix $\mathbf{W}_E \in \mathbb{R}^{d \times h}$ and bias term $\mathbf{b}_E \in \mathbb{R}^h$, while the decoder is characterized by a weight matrix $\mathbf{W}_D \in \mathbb{R}^{h \times d}$ without an additional bias term. A learned bias vector $\mathbf{b} \in \mathbb{R}^d$ is subtracted before encoding and reintroduced after decoding, leading to the following SAE formulation

$$\text{SAE}(\mathbf{x}) = (\text{bias}^{-1} \circ g \circ \text{ReLU} \circ f \circ \text{bias})(\mathbf{x}) = \mathbf{W}_D^T \, \text{ReLU}(\mathbf{W}_E^T(\mathbf{x} - \mathbf{b}) + \mathbf{b}_E) + \mathbf{b}. \tag{1}$$

Since sparsity already constrains the effective dimensionality of the concept space, the hidden dimension $h$ is chosen such that $h \gg d$. The expansion factor $\mu$ defines this relationship as $h = \mu \cdot d$, where $\mu$ is a hyperparameter.

The training objective of the SAE balances an $L_2$ reconstruction loss, preserving key information from the CLIP embeddings, with an $L_1$ sparsity penalty, ensuring concept selectivity

$$\mathcal{L}_{\text{SAE}}(\mathbf{x}) = \|\text{SAE}(\mathbf{x}) - \mathbf{x}\|_2^2 + \lambda_1 \cdot \|(\text{ReLU} \circ f)(\mathbf{x})\|_1. \tag{2}$$

Here, $\mathbf{x}$ represents image embeddings from the dataset $\mathcal{D}_{\text{extract}}$, obtained via the CLIP image encoder while ignoring text annotations. The hyperparameter $\lambda_1$ regulates the trade-off between reconstruction fidelity and sparsity. By enforcing sparsity, DN-CBM mitigates polysemanticity, ensuring that discovered concepts are more interpretable and less entangled.

### 6.1.2 Automated Concept Naming.

Following concept extraction, each neuron in the SAE's hidden representation must be assigned a meaningful label. This is accomplished by aligning discovered concepts with representations in the CLIP embedding space.

To assign interpretable labels to the hidden neurons of the SAE, a predefined vocabulary $\mathcal{V}$ is used. Each word $v \in \mathcal{V}$ is mapped to a CLIP embedding via the text encoder, forming a set of vocabulary embeddings $\mathcal{E} = \{\mathcal{T}(v)\}_{v \in \mathcal{V}}$. The standard basis vectors $\mathcal{C} = \{\mathbf{e}_1, \dots, \mathbf{e}_h\}$ in $\mathbb{R}^h$ are then passed through the decoder $g$, extracting the CLIP embedding of each neuron:

$$\mathbf{W}_D^T \mathbf{e}_n. \tag{3}$$

A name is assigned to each neuron by finding the vocabulary word whose CLIP embedding maximizes cosine similarity with the corresponding neuron embedding:

$$s_n = \arg\max_{v \in \mathcal{V}} [\cos(\mathbf{W}_D^T \mathbf{e}_n, \mathcal{T}(v))]. \tag{4}$$

The choice of vocabulary $\mathcal{V}$ is a key factor in obtaining meaningful names, as the granularity of the available words directly impacts the interpretability of the assigned labels.

### 6.1.3 Concept Bottleneck Model for Classification.

After concept discovery and naming, the extracted sparse representations serve as the foundation for a Concept Bottleneck Model (CBM). Given an input image, the SAE produces a sparse concept vector, which is then used for classification.

A linear probe $h : \mathbb{R}^h \to \mathbb{R}^c$ is introduced to map concept vectors to class logits. With $\mathcal{I}(\cdot)$ as the CLIP image encoder and $f(\cdot)$ as the SAE encoder, the full CBM model is defined as

$$t(\mathbf{x}) = (h \circ \text{ReLU} \circ f \circ \mathcal{I})(\mathbf{x}). \tag{5}$$

Training of the classifier is performed on a dataset $\mathcal{D}_{\text{extract}}$, divided into $c$ categories. The classification loss consists of a cross-entropy term and an $L_1$ penalty on the classifier weights to enhance interpretability

$$\mathcal{L}_{\text{probe}}(\mathbf{x}, y) = \text{CE}(t(\mathbf{x}), y) + \lambda_2 \|\omega\|_1. \tag{6}$$

Here, $\lambda_2$ is the sparsity coefficient, and $\omega$ represents the weights of the linear probe. The image encoder $\mathcal{I}(\cdot)$ and the SAE encoder $f(\cdot)$ remain frozen during training, ensuring that only the classifier is optimized.

The embeddings from both RN50 and ViT-B/16 reside in the same CLIP space, facilitating direct alignment between image and text embeddings for downstream classification tasks.

## 6.2 Penalizing spurious correlations with modified loss

Our training objective combines three components: a standard *classification loss* (to learn discriminative features), a *sparsity loss* (to encourage sparse usage of concepts), and our key *penalty loss* (to intervene on concept usage and combat spurious correlations). Formally, for each batch of training samples $(\mathbf{x}, y)$, the total loss is:

$$\mathcal{L}_{\text{total}}(\mathbf{w}) = \mathcal{L}_{\text{class}}(\mathbf{w}, \mathbf{x}, y) + \lambda_{\text{spar}} \mathcal{L}_{\text{spar}}(\mathbf{w}) + \lambda_{\text{penalty}} \mathcal{L}_{\text{penalty}}(\mathbf{w}, y), \tag{7}$$

where $\mathcal{L}_{\text{class}}$ is a cross-entropy term ensuring correct classification of input $\mathbf{x}$, $\lambda_{\text{spar}}$ is a weight for the L1 regularization on weights $\mathbf{w}$, and $\mathcal{L}_{\text{penalty}}$ enforces higher weights for concepts more similar to the target class name, while suppressing weights for concepts with lower similarity.

### 6.2.1 Penalty Loss Computation

The penalty loss is designed to encourage the model to assign higher weights to concepts that are semantically aligned with the target class while suppressing weights assigned to irrelevant concepts. This is achieved by incorporating three key components: (i) penalizing large weights on concepts with low similarity to the target class, (ii) encouraging alignment between the learned weight distribution and the similarity distribution, and (iii) enforcing non-negative weights for relevant concepts.

**Mathematical Formulation.** Given a linear model with weight matrix $\mathbf{W} \in \mathbb{R}^{C \times D}$, where $C$ is the number of classes and $D$ is the number of concepts, the penalty loss is computed per batch of training samples. For each batch, we define:

- $\mathbf{R} \in \{0, 1\}^{C \times D}$: a binary mask indicating relevant concepts for each class,

- $\mathbf{S} \in [0, 1]^{C \times D}$: a similarity matrix quantifying how strongly each concept aligns with the corresponding class,

- $\mathbf{W}_y \in \mathbb{R}^D$: the row of the weight matrix corresponding to the ground-truth class $y$.

The penalty loss is then computed using three main terms:

**(1) Low-Similarity Penalty.** To discourage reliance on concepts that are weakly related to the class, we apply a penalty proportional to the absolute weight values of such concepts. This term is given by:

$$\mathcal{L}_{\text{low-sim}} = \sum_{i=1}^{D} |(\mathbf{W}_y)_i| \cdot \mathbf{R}_{y,i} \cdot (1 - \mathbf{S}_{y,i}). \tag{8}$$

Concepts with high similarity ($\mathbf{S}_{y,i} \approx 1$) contribute little to this penalty, while those with low similarity ($\mathbf{S}_{y,i} \approx 0$) are strongly penalized.

**(2) Cosine Similarity Loss.** To further enforce alignment between the learned weight distribution and the similarity scores, we encourage the cosine similarity between $\mathbf{W}_y$ and $\mathbf{S}_y$:

$$\mathcal{L}_{\text{cosine}} = -\frac{\mathbf{W}_y \cdot \mathbf{S}_y}{\|\mathbf{W}_y\|_2 \|\mathbf{S}_y\|_2}. \tag{9}$$

Since cosine similarity ranges from $[-1, 1]$, minimizing the negative term ensures that $\mathbf{W}_y$ and $\mathbf{S}_y$ are closely aligned in direction. This term aims to make the weights vector for each class to be in the same direction (proportionally similar) to the similarity vector between class names and concepts.

**(3) Positivity Constraint.** Since relevant concepts should have positive contributions, we introduce a penalty for negative weights assigned to relevant concepts:

$$\mathcal{L}_{\text{positivity}} = \sum_{i=1}^{D} \max(0, -(\mathbf{W}_y)_i) \cdot \mathbf{R}_{y,i}. \tag{10}$$

This ensures that relevant concepts (as determined by $\mathbf{R}$) are not assigned negative weights, which would contradict their expected contribution.

### 6.3 Additional Qualitative results

In the paper from Rao et al. they provided additional qualitative results. Our study aims to validate the claims of DN-CBM through rigorous qualitative and quantitative evaluation. We investigate task-agnostic concept discovery by identifying shared and dataset-specific concepts across CIFAR-10, CIFAR-100, ImageNet, and Places365, assessing how well DN-CBM captures semantically meaningful features in a task-agnostic setting. We analyze semantic consistency through meta-clustering, revealing how images are organized into coherent high-level structures based on concept activations. To further evaluate interpretability, we generate local and global explanations, highlighting the most influential concepts at both the individual and class levels. Examples of task agnosticity will be shown in section A1, meta-clusters in A2, local explanations in A3 and global explanations in A4.

### A1 Task-Agnostic Concept Discovery

This section examines how interpretable concepts emerge across multiple datasets in a task-agnostic manner. Rather than restricting the analysis to a single dataset, the goal is to identify concepts that generalize across CIFAR-10, CIFAR-100, ImageNet, and Places365, while also recognizing dataset-specific variations.

To achieve this, the most influential concepts are extracted and ranked using CLIP-derived embeddings. The top-K representative images for each concept are selected based on their activation strengths, allowing for a qualitative assessment of how these concepts manifest across different domains.

Figures 3 and 4 illustrate the task-agnostic nature of the discovered concepts by visualizing both high-level and low-level concepts alongside their most activating images. High-level concepts, such as *Namibia*, *Firefighter*, and *Alcoholic*, encapsulate complex semantic themes that persist across datasets, demonstrating their robustness despite visual variability.

In contrast, low-level concepts, such as *Plaid*, *Turquoise*, and *Stripes*, correspond to fundamental visual attributes, including textures, colors, and structural patterns. These concepts appear consistently across datasets, highlighting how the model organizes lower-level information independently of object identity.

### A2 Meta-Clusters

For this part we grouped images based on shared concept activations, revealing high-level thematic structures beyond individual class labels. By applying K-Means clustering to concept representations, we identified common patterns across images, regardless of their ground truth categories.

For instance, in Figure 5, we observe three distinct meta-clusters: A horse-riding cluster, characterized by concepts such as *equestrian*, *horseback*, and *jumps*. A baseball cluster, where images share concepts like

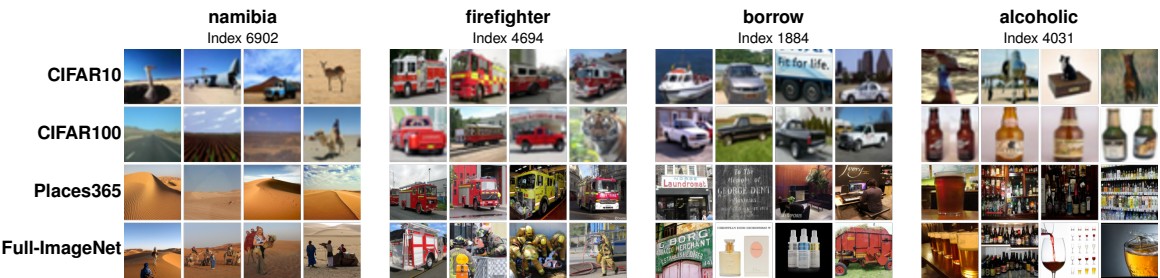

Figure 3: Visualization of high-level concepts such as "Namibia," "Firefighter," "Borrow," and "Alcoholic" across CIFAR-10, CIFAR-100, Places365, and ImageNet.

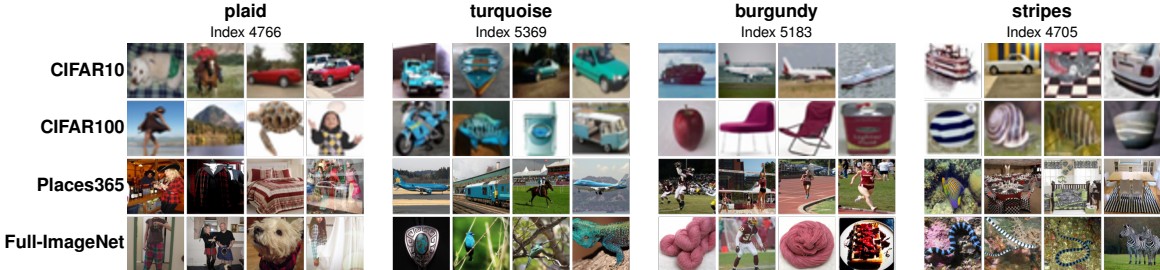

Figure 4: Visualization of low-level concepts such as "Plaid," "Turquoise," "Burgundy," and "Stripes" across CIFAR-10, CIFAR-100, Places365, and ImageNet.

*batting*, *pitching*, and *sophomore*. A snow-themed cluster, dominated by concepts such as *snowboarding*, *snowy*, and *buried*.

These clusters highlight how the model organizes knowledge, grouping semantically similar images even when they belong to different classes. This provides insight into coherent representations as well as potential spurious correlations, resulting from overreliance on background context rather than object-specific features.

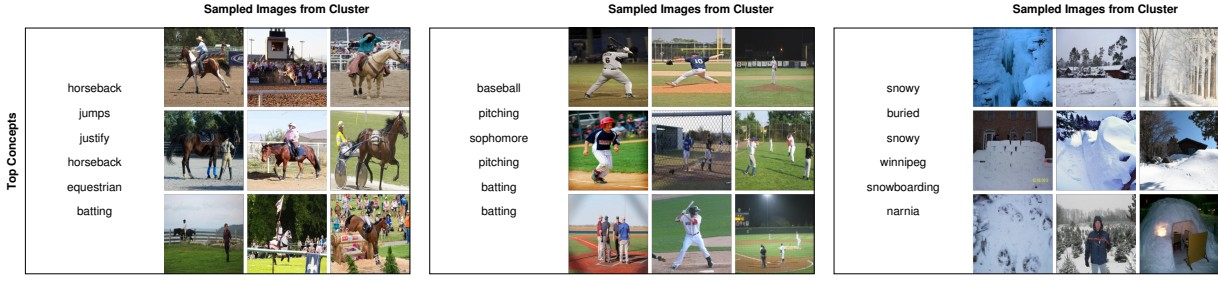

Figure 5: Examples of classes from the Places365 dataset with the top contributing concepts.

**A3 Local Explanations**

In this section, we present examples of local explanations generated by the reproduced DN-CBM on the Places365 and ImageNet datasets, using CLIP ResNet-50. For a given image, we extract its concept activations. These activations are then multiplied by the classifier weights to determine the contribution of each concept to the predicted class. The concept contributions are ranked, and we extract the top-K most influential concepts for each image. Each image is presented alongside its predicted class, ground truth label, and a bar chart displaying the top contributing concepts with their corresponding strengths. In Figure 6a the model correctly identified the *overskirt* by associating it with related fashion concepts such as *gowns*

*skirt*, and *dress*. This suggests that the model successfully captures semantic relationships between concepts, allowing it to leverage contextual cues to enhance classification accuracy.

On the other hand the model misclassified a *shed* as a *ski resort*. This error is likely due to the presence of contextual elements such as **snow** and other winter-related cues (e.g., *sauna*, *Andorra*, *Alps*). This suggests that the model sometimes relies too much on **scene context** rather than the object's intrinsic properties, leading to systematic misclassifications in certain environments — an example of a spurious correlation.

Overall, the explanations effectively capture the key characteristics of the input images, exhibit significant diversity, and provide insights into the model's misclassified decisions.

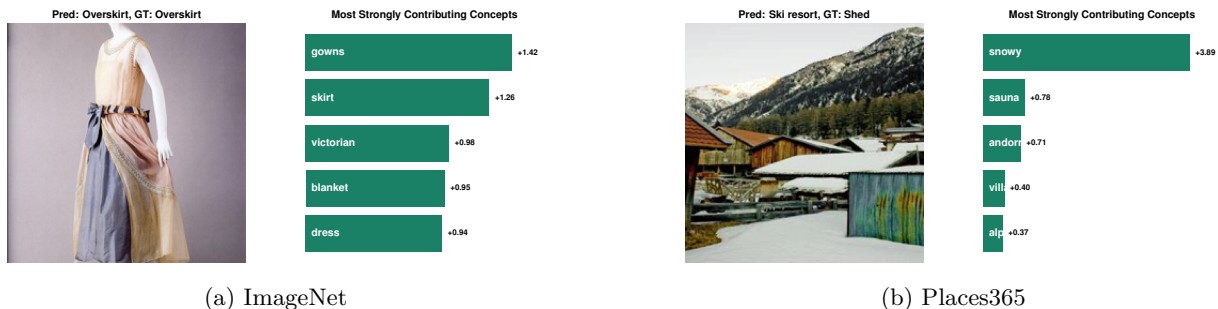

(a) ImageNet    (b) Places365

Figure 6: Explaining decisions using CBM

## A4 Global Explanations

In this section, we analyze global explanations by identifying and visualizing the most influential concepts associated with each class in the Places365 and ImageNet datasets. Unlike local explanations, which focus on individual image-level interpretations, global explanations provide an overview of class-level concept associations, helping us understand the broader patterns that the model relies on when making predictions.

To generate these global explanations, we extract concept activations across all images belonging to a specific class and compute the average contribution of each concept. The most influential concepts for each class are ranked and presented alongside representative images from the dataset.

For example, in Figure 8, the model identifies a *shopping mall indoor* scene, with the most relevant concepts including *conventions*, *theaters*, *shopping*, and *malls*. This suggests that the model strongly associates shopping malls with large public spaces, events, and commercial environments. Similarly, for the *shoe shop* class, the top concepts include *shelves*, *collection*, *heels*, and *footwear*, indicating that the model recognizes key visual elements related to shoes.

However, in some cases, the model's concept associations show spurious correlations. For instance, looking at the grasshopper in Figure 7, the model has *caterpillar*, *parrot*, *frog*, and *insects* as top concepts. While these concepts are related to nature and small creatures, the presence of *parrot* and *frog* suggests that the model may be picking up on broad ecological similarities rather than fine-grained insect characteristics.

### 6.4   User study

In this section, we describe the details of the user study we conducted in order to quantitatively measure the consistency and accuracy of our discovered and named concepts. Our goal was to see if the trends reported by the authors were reproducible on a different set of users.

**Node Selection.**   Following the paper, we sorted the SAE nodes for the CLIP ResNet-50 model, based on the cosine similarity with the text embedding vector of the name assigned to them. We then uniformly at random sample nodes from three bins where the alignment is the highest, intermediate, and lowest. From the concept space including 8129 nodes, the high bin included the top 2000 nodes, the low bin included the 2000 less aligned nodes and the intermediate contained the remaining 4192. From each bin we randomly

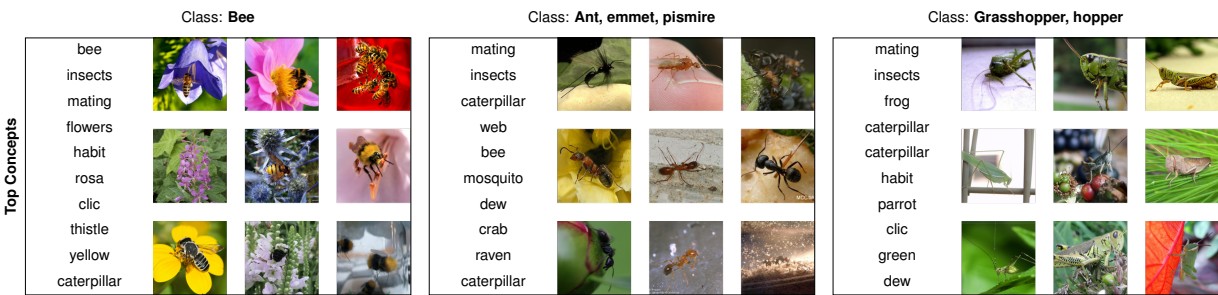

Figure 7: Global explanations from ImageNet dataset

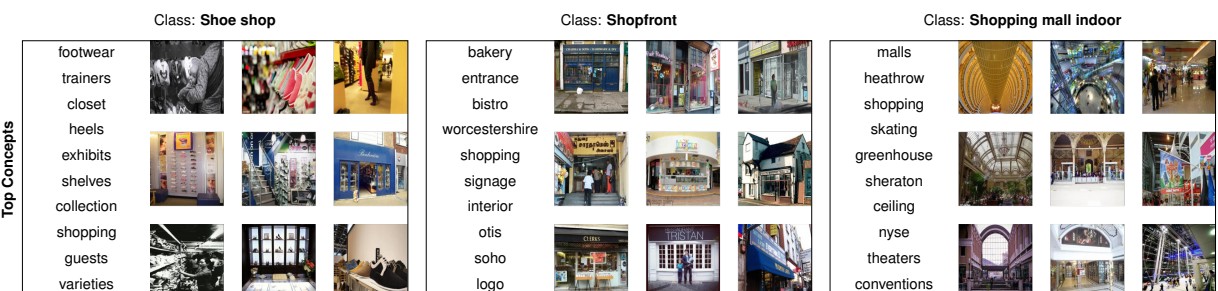

Figure 8: Global explanations from Places365 dataset

sampled 10 concepts, and then chose 5 that are commonly used in everyday language. The 20K Vocabulary includes multiple words that are not as common in colloquial English (for example the word "bespectacled" appeared in the top 12 most aligned concepts), therefore sampling only randomly would have most likely caused the introduction of unfamiliar words for some users and a biased response in the survey. This manual check was not mentioned in the authors' analysis but in order to have an accurate representation of the users' opinions, we decided to add it.

**Question Structure.** For each node, we extracted the top four activating images from three diverse datasets – ImageNet, Places365, and CIFAR100 – and create a grid of twelve images. For each set of images we also provided the word that corresponded to the name assigned to the node. We asked the same questions that were proposed by the authors: (1) *Does the set of images correspond to a human-understandable concept?* (2) *Does the word provided with the set of images accurately describe the common concept across the images?*. For each question, the participants can answer on a 5-point scale, rating their agreement from "Strongly Disagree" to "Strongly Agree".

**Survey Response.** We randomly ordered the 15 nodes in the survey and we decided to repeat one to check how stable and coherent would this human quantitative analysis be. Before starting, we provided the participants with two examples, as the authors did, to help them familiarize with the task. We published the survey internally, and received 26 responses. The trends that we obtained are in line with the ones obtained by the authors, as it is shown in Figure 9. Furthermore, by analyzing the responses from the duplicated concept set inserted in the survey, we can observe that 3 users out of 26 lowered their score for the concept consistency and two of them increased it, when encountering the same concept a second time. Regarding the name accuracy, during the second interaction with the same concept, 4 users increased their score, while 3 of them lowered it. As we can notice, the user responses are not always coherent, but the changes in scoring compensate each other, so overall the average results coming from the two identical nodes are very similar. In this way we proved that the the user survey approach lead to consistent results, even if the singular responses of the participants are not always coherent.

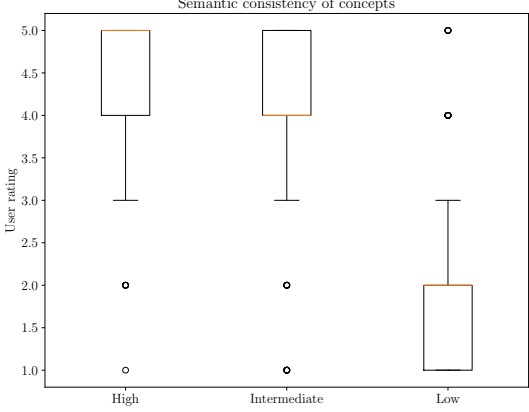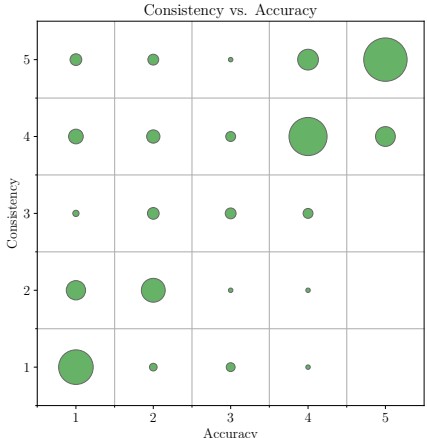

Figure 9: **User study on concept accuracy.** Left: The trend of semantic consistency follows the distribution of the three sets of alignment with the text embeddings of the concept names. Right: We plot the scores for semantic consistency against name accuracy from human evaluators. Our results are comparable to the ones obtained by the authors.

### 6.5 Understanding the concept space

In this section we provide further comments on our study regarding concreteness and polysemanticity in the learned concept space. In Figure 10 we present the correlation scores and plots between concreteness and concept alignment, and conclude that for CLIP ViT/B-16 there is no apparent correlation between these two quantities while for CLIP ResNet-50 there is a positive correlation in the direction of concrete concepts being more aligned. This finding is here presented to support the fact that the increase on human understandability that arises from learning more concrete concepts is not counteracted by a great misalignment in the concepts. In fact, it is further increased for CLIP ResNet-50 given the slight positive correlation.

Moreover, in Figure 11 we see that the average Synset count (word meanings) in WordNet, which is a direct measure of polysemanticity, generally increases during training. This shows that even though polysemanticity is not fundamental for the downstream task it is indeed necessary for the reconstruction objective of the SAE.

### 6.6 Concept Intervention

In this section we report some additional results concerning the intervention analysis on the Waterbirds100 dataset, measuring the impact of retaining or removing key concepts on classification accuracy.

To systematically analyze the role of concept selection, we create three versions of the linear probe:

- **Standard Probe:** No intervention, uses all concepts for inference.

- **Bird-Only Probe:** Only bird-related concepts out of the top 10 are retained, with non-bird concepts removed.

- **No-Bird Probe:** Bird-related concepts are removed, retaining only non-bird concepts in the top 10.

The concepts that were zeroed out for each probe variant are shown in Table 9.

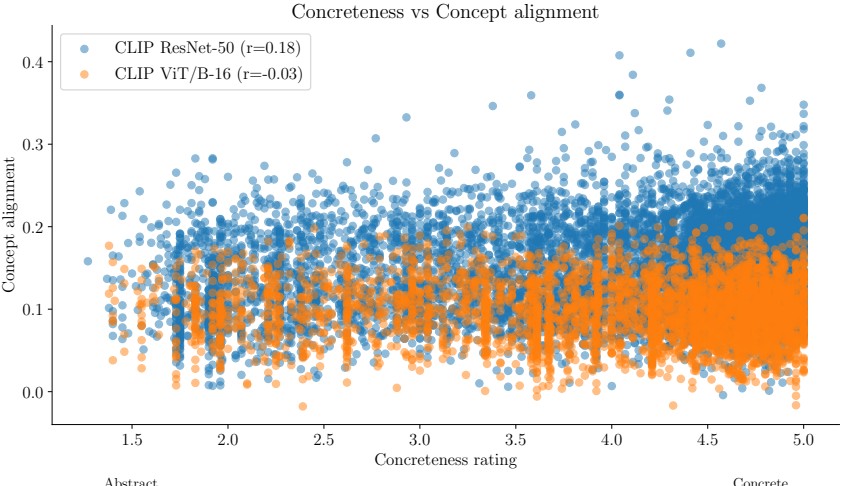

Figure 10: Relation between concreteness rating (ranging from abstract to concrete) and concept alignment for two CLIP models: CLIP ResNet-50 (blue) and CLIP ViT/B-16 (orange). The correlation coefficient (r) indicates a weak positive correlation (r=0.18) for CLIP ResNet-50, while CLIP ViT/B-16 exhibits almost no correlation (r=-0.03). However, note that there is a high imbalance in the number of points in the concrete spectrum as compared to the abstract spectrum.

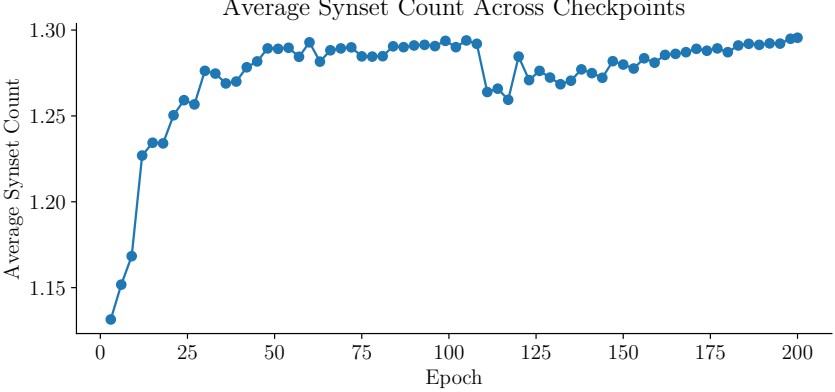

Figure 11: Evolution of polysemanticity throughout training of the SAE for CLIP ResNet-50. In the x-axis we have the number of epochs, ranging from 3 to 200 and with measures every three epochs. In the y-axis we have the average synset of the words that were assigned to the particular concept space of that give epoch. We see that in general terms, it increases with the number of epochs.

Table 8: Top 10 concepts identified by the linear probe for each class.

| Landbird | | Waterbird | |
|---|---|---|---|
| **Concept** | **Weight** | **Concept** | **Weight** |
| Rainforest | 15.2224 | Ducks | 9.6791 |
| Magnolia | 4.1347 | Thames | 6.1616 |
| Parrot | 4.0557 | Beach | 4.9798 |
| Grass | 2.8577 | Seals | 3.0335 |
| Sparrow | 2.5934 | Canoeing | 2.9707 |
| Owl | 2.4753 | Flying | 0.8073 |
| Clic | 0.2446 | Yacht | 0.2091 |
| Social | 0.2023 | Casper | 0.0969 |
| Bild | 0.1307 | Anguilla | 0.0833 |
| Eagle | 0.1129 | Gazette | 0.0656 |

Table 9: Concepts zeroed out for each intervention type.

| Probe Variant | Zeroed Concepts |
|---|---|
| Bird-Only | Rainforest, Magnolia, Grass, Clic, Social, Bild, Thames, Beach, Seals, Canoeing, Flying, Yacht, Casper, Anguilla, Gazette |
| No-Bird | Parrot, Sparrow, Owl, Eagle, Ducks |

