# OpenReview forum: "Reproducibility Study of "Discover-then-Name: Task-Agnostic Concept Bottlenecks via Automated Concept Discovery""
_TMLR — Rejected by TMLR_

### Review · Reviewer_y5Yu · 2025-03-12

**Summary Of Contributions:**

The authors study the re-producibility of "Discover-then-Name: TaskAgnostic Concept Bottlenecks via Automated Concept Discovery" (Rao et al.). The autors replicate the experiments of the original paper using public available github code, make minor changes to reproduce the original experiments. The authors show that they can replicate the experiments and conclusions of the DN-CBM. The authors also conduct an intervention analysis, concluding that retaining only class related concepts can significantly improve overall accuracy, while removing these concepts lead to a substantial drop in performance.

**Audience:**

No

**Broader Impact Concerns:**

No broader impact concerns.

**Claims And Evidence:**

Yes

**Requested Changes:**

- Provide more in-depth analysis on, e.g., concept intervention, automatic concept discovery using more advanced model such as LLMs or world models. The current paper provides only minor/trivial insights.

- Conduct thorough literature reviews of previous studies, clearly delineate how DN-CBM differs from these, and highlight the novel aspects of the DN-CBM models.

In general, the discussion of related works, comparison with previous baselines, and the extended experiments require significant additional effort and currently fall short of the accept criteria required by TMLR.

**Strengths And Weaknesses:**

Strengths:

- the paper is easy to follow, with good presenting and explanation about each steps of the experiments.
- Conclude that automatic intervention methods would underperform when comparing to manual human intervention (but this sounds trivial since human intervention are always better for the most tasks)
- Provide sufficient explanations on for the original paper.

There are **two main weakness** of the paper:
- First, there are minor contributions/novel findings that were revealed in this paper comparing with the original paper. The replicated experiments or the extended analysis on human intervention do not offer novel insights about the concept-based neural networks, which fail to the criteria of TMLR (https://jmlr.org/tmlr/acceptance-criteria.html): "A machine learning class report that re-runs the experiments of a published paper has educational value to the students involved. But if it doesn't surface generalizable insights, it is unlikely to be of interest to (even a subset of) the TMLR audience.".

- Second, both this paper and the original work by Rao et al. lack comprehensive literature reviews. For instance, an earlier study titled "CEIR: Concept-based Explainable Image Representation Learning" by Cui et al., which made its paper and code publicly available at least a year prior to DN-CBM (Rao et al.), already introduced a almost the same idea and framework for concept discovery. Yet, neither this paper nor DN-CBM (Rao et al.) acknowledges or compares with this prior work. Notably, Cui et al. suggested utilizing LLMs to provide predefined concept sets, which represents a more advanced and flexible approach compared to the fixed concept pools used by CLIP-Dissect (Oikarinen et al.).

References:
- Rao, Sukrut, et al. "Discover-then-name: Task-agnostic concept bottlenecks via automated concept discovery." European Conference on Computer Vision. Cham: Springer Nature Switzerland, 2024.
- Cui, Yan, et al. "Ceir: Concept-based explainable image representation learning." arXiv preprint arXiv:2312.10747 (2023).
- Oikarinen, Tuomas, and Tsui-Wei Weng. "Clip-dissect: Automatic description of neuron representations in deep vision networks." arXiv preprint arXiv:2204.10965 (2022).

---

> ### Author Response · Authors · 2025-04-12
> **Answer to Reviewer y5Yu**
>
> > First, there are minor contributions/novel findings that were revealed in this paper comparing with the original paper. [...] which fail to the criteria of TMLR […]
>
> Thank you for pointing this out. We’d like to respectfully clarify that the citation from the TMLR acceptance criteria appears to be incomplete. The full context reads:
> “A machine learning class report that re-runs the experiments of a published paper has educational value to the students involved. But if it doesn't surface generalizable insights, it is unlikely to be of interest to (even a subset of) the TMLR audience.
> On the other hand, a proper reproducibility report that systematically studies the robustness or generalizability of a published method and lays out actionable lessons for its audience could satisfy this criterion.”
>
> We believe our work clearly aligns with the second category. While it is grounded in reproducing the DN-CBM framework, our study surfaces several generalizable and actionable insights:
> - Concept space analysis: We introduced a novel analysis of concept concreteness and polysemanticity, demonstrating that the SAE tends to learn more concrete and monosemantic concepts—key features for interpretability.
> - Modified loss function: We designed a new loss function that significantly improves classification performance on the biased Waterbirds-100 dataset by reducing reliance on spurious correlations (which is a known problem in deep learning image classification) and leads to more semantically relevant concepts on CIFAR-10, as confirmed by a user study.
> - Automatic concept intervention: We developed and evaluated a fully automated concept intervention strategy, offering an alternative to manual selection and shedding light on the limitations of embedding-based alignment for this task.
>
> > Provide more in-depth analysis on, e.g., concept intervention, automatic concept discovery using more advanced model such as LLMs or world models.
>
> We are really interested in the suggestion to use world models. We also think that this would require a proper formalization of world models for this type of tasks, which is non trivial, and it would be out of the scope of a reproducibility paper of a concept bottleneck model.
> As for the use of LLMs for concept naming, we intentionally avoided them to remain faithful to the original DN-CBM setup (as is expected in a reproducibility study) and to maintain consistency with our focus on fairness and transparency. LLM-based naming introduces additional opacity and significant computational cost, which can hinder both interpretability and accessibility—especially in research settings concerned with equity and reproducibility.
> We hope this addresses your concern and clarifies the broader value of our contributions.
>
> > Second, both this paper and the original work by Rao et al. lack comprehensive literature reviews. For instance, an earlier study titled "CEIR: Concept-based Explainable Image Representation Learning" by Cui et al., which made its paper and code publicly available at least a year prior to DN-CBM (Rao et al.), already introduced a almost the same idea and framework for concept discovery. Yet, neither this paper nor DN-CBM (Rao et al.) acknowledges or compares with this prior work. Notably, Cui et al. suggested utilizing LLMs to provide predefined concept sets, which represents a more advanced and flexible approach compared to the fixed concept pools used by CLIP-Dissect (Oikarinen et al.).
>
> > Conduct thorough literature reviews of previous studies, clearly delineate how DN-CBM differs from these, and highlight the novel aspects of the DN-CBM models.
>
> Thank you very much for this feedback. In the introduction we included a motivation for why we chose this specific paper and the novel findings that are presented. Moreover, in the same added paragraph we also cite CEIR that was missing in the literature of the original paper. There we highlight the differences between the two papers and we show the benefits of DN-CBM compared to CEIR.
> In particular, this study focuses on DN-CBM due to its intrinsic task-agnostic nature, which distinguishes it from models such as CEIR. In DN-CBM, the sparse autoencoder is trained once and remains fixed, thereby necessitating only the retraining of a linear probe for adapting to new tasks, making this model more computationally advantageous. Furthermore, DN-CBM employs a fixed (but possibly tunable) vocabulary based on unigrams rather than relying on expensive large language models for generating task-specific concepts. This design not only reduces computational overhead but also enhances reproducibility and ease of integration into various downstream applications. Additional advantages include the simplicity of the training procedure, good performances on complex datasets such as ImageNet and improved efficiency, which together render DN-CBM a compelling candidate for further empirical validation.”

---

### Review · Reviewer_d3sA · 2025-03-20

**Summary Of Contributions:**

This is a reproducibility study paper. I am not sure this kind of reproducibility study falls in the scope of TMLR, but I checked the official Acceptance Criteria, it seems ok.

The target of reproducibility is the DN-CBM framework, proposed by Rao et al., advances concept-based interpretability by leveraging Sparse Autoencoders (SAEs) to automatically discover and name meaningful concepts without predefined dictionaries. In this study, authors  replicate DN-CBM’s core findings, confirming its ability to extract semantically meaningful concepts while maintaining competitive classification performance on datasets like ImageNet, Places365, CIFAR-10, and CIFAR-100, and effectively clustering related concepts in the latent space.

Further, extensions reveal that discovered concepts are more concrete and monosemantic, with polysemantic concepts having minimal classification impact. A novel loss function improves accuracy by reducing reliance on spurious cues, validated through intervention analysis on Waterbirds100. While automatic concept intervention offers an alternative, human selection remains more effective. The findings verified by this study will benefit the research community of interpretable deep learning.

Overall, this is a good paper with clear writing and solid results. I believe lots of audience will appreciate this paper.

**Audience:**

Yes

**Broader Impact Concerns:**

No concerns. This paper could have a good impact in the community.

**Claims And Evidence:**

Yes

**Requested Changes:**

See above.

**Strengths And Weaknesses:**

Pros:

1.I like the way that authors summarize key claims in section 2, making it much easier to get the key points and easy to follow quickly.

2.Overall, this verification paper is well-written and easy to follow. Will be very helpful for related researchers in this field.

3.Another advantage of this paper is that, rather than purely reproduce the original “discover-then-name” paper, it further explore the SAE’s concept space. These extensions will benefit future studies in the area of DNN interpretability.

4. This paper make some further designs that improve the original paper, e.g., the modified loss function.

Questions:
1.IN section 3.5, could you explain how to get the carbon footprint? Also, is this the amount of co2 of finishing the training or taking consideration of various exploration phases.

2.The resolution of figures could be improved by using PDF format, e.g., Figure 11.

3.Section 6.5 is not easy to understand. Summarizing the findings from Fig.10 in one sentence would benefit future readers.

---

> ### Author Response · Authors · 2025-04-12
> **Answer to Reviewer d3sA**
>
> Dear Reviewer,
> Thank you for your detailed review.
> We will comment on the requested changes.
>
> > 1.In section 3.5, could you explain how to get the carbon footprint? Also, is this the amount of co2 of finishing the training or taking consideration of various exploration phases.
>
> Thank you for pointing this out. In the revised version of the manuscript, we have clarified this in Section 3.5 and included a direct reference to the Machine Learning Emissions Calculator used for the estimation Lacoste et al., 2019. The reported value reflects the total estimated emissions across all phases of the reproduction study, including both core training and exploratory runs for the extensions. We also specify the assumptions made (e.g., single GPU usage) and how this affected the final estimate.
> > 2.The resolution of figures could be improved by using PDF format, e.g., Figure 11.
>
> Thank you for your feedback, we will ensure that all figures are updated to high-resolution versions where necessary.
>
> > 3.Section 6.5 is not easy to understand. Summarizing the findings from Fig.10 in one sentence would benefit future readers.
>
> This is a very valid concern. We added further explanations of the Figures as well as their relation with Section 4.2.1. In particular, we pointed out that for CLIP ViT/B-16 there is no apparent correlation between these two quantities while for CLIP ResNet-50 there is a positive correlation in the direction of concrete concepts being more aligned. This finding is here presented to support the fact that the increase on human understandability that arises from learning more concrete concepts is not counteracted by a great misalignment in the concepts. In fact, it is further increased for CLIP ResNet-50 given the slight positive correlation.

---

### Review · Reviewer_hX7N · 2025-04-04

**Summary Of Contributions:**

The paper aims at conducting a reproducibility study of "Discover-then-Name: Task-Agnostic Concept Bottlenecks via Automated Concept Discovery" which is a paper published at ECCV 2024. The original paper focuses on addressing the ‘black-box’ problem of deep neural networks and it proposes to leverage Sparse Autoencoders (SAEs) for automatic concept discovery and naming. This paper reproduced the core findings in the original papers by using the code provided in the original paper, verifying the ability of the original paper for extracting meaningful concepts while maintaining competitive classification performance across multiple datasets. Apart from reproducing the original paper, the also also conduct extensions to show deeper insights for the interpretability and robustness of the original paper.

**Audience:**

Yes

**Broader Impact Concerns:**

The paper does not include a statement of Broader Impact. Though the paper mainly focuses on validating an existing paper, there can be some ethical implications should be considered. For example, automated naming of concepts using vocabularies like WordNet or large embeddings could unintentionally assign biased or sensitive labels, potentially leading to misconceptions.

**Claims And Evidence:**

Yes

**Requested Changes:**

1. More detailed explanation of why this paper is selected for replication and extensions such as modified loss function and automatic concept interventions.

2. As the proposed extension of a modified loss can lead to a drop in performance in datasets that does not appear in training and the number of sample in user study is relatively small, a larger-scale user study (50-100) would be better for ensuring the statistically significant and more reliable results.

**Strengths And Weaknesses:**

+ ***Strengths***

1. The paper aims at performing a reproducibility study of the paper which is published at ECCv 2024 and the original paper has achieved significant advancement in concept-based interpretability.

2. The paper also provides extensions for providing deeper insights for the interpretability and robustness of the original paper.

3. In general, the paper is clear and easy to follow.

- ***Weaknesses***

1. The replication is relatively easy as they are performed by using the implementation (Code, hyperparameters etc) provided by the original papers.

2. The paper clearly states the reason of reproducing of the original paper is that the original paper has shown significant advancement in concept-based interpretability. However, more detail on why this paper is selected for replication and extensions such as modified loss function and automatic concept interventions would strengthen the motivation.

3. The results reported in Sec. 4.2.2 shows that the introduced loss improves the performance in the Waterbirds-100 datasets while it lead to a drop in the CIFAR10 dataset which does not appear in training. This results reveals the limitations in the generalizability of their proposed extensions. Additionally, the user study is conducted on a relatively small sample (32 and 26 in this paper).

---

> ### Author Response · Authors · 2025-04-12
> **Answer to Reviewer hX7N**
>
> Dear Reviewer,
> Thank you for your detailed review.
> We will comment on the requested changes.
>
> > More detailed explanation of why this paper is selected for replication
>
> In recent tasks, LLMs obtain state-of-the-art performance in a wide variety of tasks. But this performance comes at the expense of computational power and a significant environmental impact (Strubell, E., Ganesh, A., & McCallum, A. (2020). Energy and Policy Considerations for Modern Deep Learning Research). To this regard, the most performing CBMs up to date were all based in LLMs (see baselines of the original and reproduced paper), which indicates that achieving better performance without the use of LLMs, as is the case of DN-CBMs, is a significant advancement at a fraction of the (computational and derived) cost. We incorporated this explanation into the paper.
>
> > and extensions such as modified loss function and automatic concept interventions.
> We included a motivation to why we introduced an automatic concept intervention mechanism.
>
> The main reason is that the original paper proposed a manual concept intervention mechanism that was shown to improve performance on the Waterbirds100 dataset (that has only two classes). With bigger datasets (e.g. ImageNet that accounts to 21k classes) such intervention would be infeasible. Regarding the modified loss function, we believe that it was already well motivated. The main motivation is that “deep learning models often exploit spurious correlations in training data, leading to poor generalization when these patterns do not hold in test settings”, this loss function is designed to penalize these spurious correlation in order to get a more robust model and we proved its efficacy on the Waterbirds100 dataset.
>
> > the number of sample in user study is relatively small, a larger-scale user study (50-100) would be better for ensuring the statistically significant and more reliable results.
>
> To address this comment, we broadened the sample size of our user study and collected a total of 52 responses from fellow researchers, in order to ensure statistical significance as suggested. The new obtained results are the same as the previously obtained, therefore the observations based on the user study were not modified.
>
>
> > as proposed extension of a modified loss can lead to a drop in performance in datasets that does not appear in training
>
> This is a fair concern. To clarify, the Sparse Autoencoder (SAE) is trained on the CC3M dataset, which does not include CIFAR-10 images, ensuring that the concept discovery process remains task-agnostic. For the linear probe, CIFAR-10 was used both for training and for evaluation on a held-out test partition, as described in Section 3.2.
> Regarding the observed accuracy drop on CIFAR-10 with the proposed loss: we agree that this highlights limitations in the generalizability of our extension. However, our primary goal with this loss was not solely to improve classification accuracy, but to enhance interpretability and robustness—particularly in scenarios with spurious correlations, as demonstrated by the performance improvements on the Waterbirds-100 dataset.
> It is worth noting that CIFAR-10 is not designed to exhibit strong spurious correlations, while Waterbirds-100 explicitly contains such biases (e.g., background–label associations). The effectiveness of our loss function on Waterbirds-100 suggests that it may generalize well to other biased datasets, where guiding the model away from confounding features is more beneficial. In contrast, in unbiased settings like CIFAR-10, where there are fewer spurious correlations to mitigate, the benefits of the penalty loss are less pronounced.
> Finally, while our initial user study on CIFAR-10 involved 26 participants, we have since increased the sample size to 52. The updated results reinforce our earlier findings: despite the slight drop in accuracy, participants consistently judged the concepts surfaced by the modified loss as more semantically aligned with class labels, suggesting an improvement in model transparency.
>
> > Though the paper mainly focuses on validating an existing paper, there can be some ethical implications should be considered. For example, automated naming of concepts using vocabularies like WordNet or large embeddings could unintentionally assign biased or sensitive labels, potentially leading to misconceptions.
>
> Thank you for pointing this out. It is indeed a very valid concern and will be included.

---

### Decision · Action_Editor_GXGz · 2025-05-20

**Recommendation:** Reject

**Comment:**

Reviewer y5Yu pointed out that the paper lacks generalizable insights, and thus does not satisfy the TMLR criteria.  The authors are recommended to follow the following suggestions made by the reviewer in response to the rebuttal:

- No additional analysis was provided regarding the suggested concept intervention or automatic concept discovery. As a result, the work still lacks clear technical contributions. Although the authors mention these directions as future work, it is strongly recommended to include more in-depth experiments and analyses rather than reiterating findings already shown in previous studies.

- The discussion of the previous work CEIR (Cui et al.) is incorrect. CEIR is also a task-agnostic framework, similar and earlier than DN-CBM (Rao et al.). The authors neither conduct quantitative or qualitative comparisons nor discuss their respective merits and limitations, even though both works are open-sourced and there should be no barrier to reproducibility.

**Audience:**

As Reviewer y5Yu pointed out, the paper lacks generalizable insights.

**Claims And Evidence:**

All reviewers agree that the claims are supported.

**Resubmission Of Major Revision:**

The authors may consider submitting a major revision at a later time.